# Online Multi-Contact Motion Replanning for Humanoid Robots with Semantic 3D Voxel Mapping: ExOctomap

**DOI:** 10.3390/s23218837

**Published:** 2023-10-30

**Authors:** Masato Tsuru, Adrien Escande, Iori Kumagai, Masaki Murooka, Kensuke Harada

**Affiliations:** 1Graduate School of Engineering and Science, Osaka University, Toyonaka 560-8531, Japan; 2CNRS-AIST Joint Robotics Laboratory, International Research Laboratory, National Institute of Advanced Industrial Science and Technology (AIST), Tsukuba 305-8560, Japan; 3INRIA Grenoble Rhône-Alpes, 38330 Montbonnot-Saint-Martin, France; 4Automation Research Team, Industrial Cyber-Physical Systems Research Center, National Institute of Advanced Industrial Science and Technology (AIST), Tokyo 135-0064, Japan

**Keywords:** humanoid robot, multi-contact, motion planning, voxel mapping

## Abstract

This study introduces a rapid motion-replanning technique driven by a semantic 3D voxel mapping system, essential for humanoid robots to autonomously navigate unknown territories through online environmental sensing. Addressing the challenges posed by the conventional approach based on polygon mesh or primitive extraction for mapping, we adopt semantic voxel mapping, utilizing our innovative Extended-Octomap (ExOctomap). This structure archives environmental normal vectors, outcomes of Euclidean Cluster Extraction, and principal component analysis within an Octree structure, facilitating an *O* (log *N*) efficiency in semantic accessibility from a position query $x\in R^{3}$. This strategy reduces the 6D contact pose search to simple 3D grid sampling. Moreover, voxel representation enables the search of collision-free trajectories online. Through experimental validation based on simulations and real robotic experiments, we demonstrate that our framework can efficiently adapt multi-contact motions across diverse environments, achieving near real-time planning speeds that range from 13.8 ms to 115.7 ms per contact.

## 1. Introduction

Humanoid robots are tasked with navigating environments characterized by uncertainty and dynamic properties, including disaster-stricken areas, radioactive zones, and construction sites. In such contexts, robots often require multi-contact motions, such as grasping handrails or pushing walls during ambulation.

Previous research studies [1,2,3] have primarily focused on identifying stable limb configurations and viable transitions between them, assuming a pre-mapped, static, and perfectly known environment. However, in dynamic and uncertain settings, as mentioned earlier, preparing static environmental models before robot deployment becomes impractical. As a result, humanoid robots must perceive their immediate surroundings in real time and adjust their movements accordingly. This real-time adaptation presents a significant challenge for motion planners, especially when environmental data are not adequately represented.

A comprehensive representation involving numerous polygons increases the complexity for motion planners, leading to a combinatorial explosion when identifying adjacent polygons at contact points or formulating collision-free trajectories. Such previous approaches based on graph search and optimization have faced challenges in identifying feasible contact points and have often fallen short in real-world applications. In contrast, our approach introduces the new idea that semantic voxels can provide motion planners with feasible 6D contact poses and also be useful for collision-free trajectory planning, as demonstrated in Figure 1.

For expedited multi-contact motion planning, we present “Extended-Octomap” (ExOctomap), a rapid semantic 3D voxel mapping system independent of 3D model registration. As illustrated in Figure 2, this system segments the entire environment into individual units. Beyond mere occupancy storage as seen in the original Octomap [4], ExOctomap encompasses semantic data within each voxel, enhancing the efficacy and speed of motion generation by narrowing the search space. Each voxel, structured via Octree, harbors diverse semantic data at its discretized position, as depicted in Figure 3. This attribute facilitates global shape feature acquisition at any $\left(x,y,z\right)\in R^{3}$ coordinate with an *O*(log *N*) efficiency, streamlining the determination of space occupancy, normal vectors, primitive shape types, and principal component analysis (PCA) outcomes. Subsequently, the ExOctomap, which encapsulates these semantics, is fed into our motion planner. Figure 4 provides an overview of our system architecture. Although our method begins with a set of reference contacts, these may not be conducive to actual environments. Therefore, at each step, our motion replanning system swiftly identifies subsequent feasible contacts by leveraging the ExOctomap. This capability allows for online adaptation of robot motions to the actual environment.

In contrast to past studies [1,3,5], which required recurrent planar selections to ascertain 6D contacts, our motion planner leverages the ExOctomap’s semantic integration to directly create 6D contacts through straightforward 3D grid sampling. Furthermore, this approach excels at devising end-effector trajectories devoid of collisions, which is enabled by ExOctomap’s retention of the environment’s original 3D structure for comparative analysis with voxelized trajectories. Following this, the Quadratic Programming (QP) Inverse Kinematics controller promptly computes all joint angles respecting kinematic constraints within a 2 ms cycle, enhancing the robot’s reachability through whole-body optimization and liberating the root link (torso) from specified 6D postures. This systemic structure eradicates the prerequisite of offline joint angle resolutions from the root link to the end-effectors.

This study pioneers online motion planning for humanoid robots in unknown unknown terrains. We advocate for a semantic 3D voxel mapping system to enhance congruence with 3D sampling processes and collision assessments during motion planning. The significant contributions of this study are as follows:The innovative concept of semantic storage in voxels utilizing an Octree structure.The augmentation of contact generation speed by reducing the 6D search space to either 3D or 4D through the application of semantics derived from vision recognition.The empirical substantiation of these concepts through real robot experimentation.

Our humanoid robot, HRP5P, demonstrates proficient block climbing and collision avoidance by utilizing online 3D mapping, without given 3D models.

**Figure 2 sensors-23-08837-f002:**
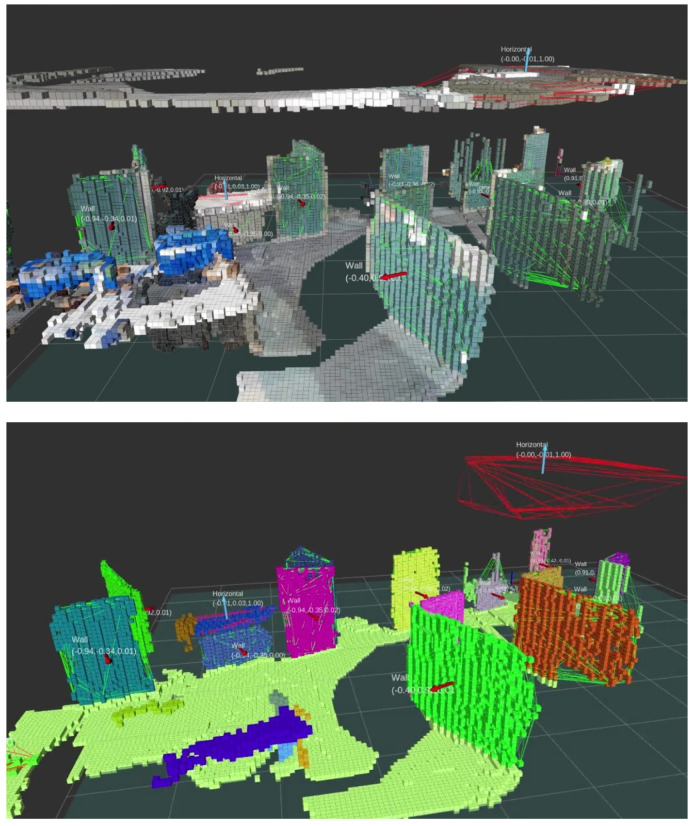
Example of ExOctomap in an indoor environment. The AzureKinect and RTabMap [6] (ver.0.20.20), an open source 3D-SLAM software for camera’s self localization are used. The resolution of the voxels is set at 3 cm. The entire area measures approximately 8.0 m × 10.0 m, featuring 959,475 voxels, which corresponds to 237.6 MB of data. For visualization purposes, we randomly change the colors for each segmented instance. The small arrows indicate the estimated normal vector of each cluster. In this demonstration, the voxel segmentation process averages around 322 ms in duration.

**Figure 3 sensors-23-08837-f003:**
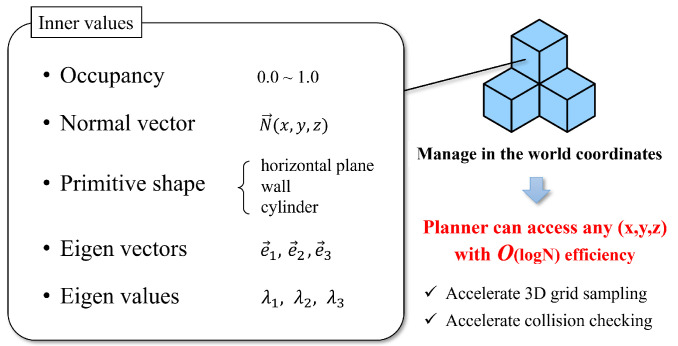
The concept of ExOctomap. Each voxel contains specific information. Similar to the original Octomap, all voxels are stored in a uniform Octree graph. It allows motion planners to access these semantics from a coordinate query (x,y,z) with *O*(log *N*) speed.

**Figure 4 sensors-23-08837-f004:**
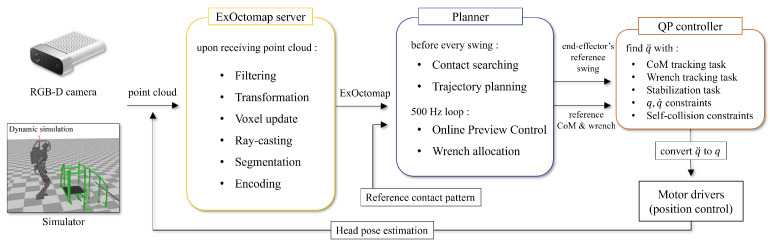
The overview of our system. The input is point cloud stream and a reference contact pattern as an indicator of the robot’s direction of movement.

## 2. Related Research

Historically, multi-contact motion planning for humanoid robots predominantly involved combinatorial searching in predefined environments, which were processed offline. Renowned works, including those by S. Tonneau et al. [1] and I. Kumagai et al. [3], presented strategies that relied on determining reachable global trajectories for the root link before pinpointing viable contact configurations for each limb with quasi-static transitions. These methodologies assumed a complete understanding and modeling of the environment prior to the robots’ ambulation.

In contrast, recent advancements have shifted toward online strategies, emphasizing footstep planning guided by the robot’s sensorial feedback. Initiatives led by J. Kuffner et al. [7,8] broke ground by introducing an online footstep planner that utilizes a 2D grid map, a concept later refined by A. Hornung et al. [9,10] and M. Marcell et al. [11] to overcome heuristic challenges encountered in A* search methodologies.

Despite their ingenuity, the confinement to 2D maps restricts robots to flat terrains, prompting the development of 3D footstep planning methodologies that leveraged 2.5D height maps and 3D planar extraction for a richer spatial representation. S. Bertrand et al. [12] and R.J. Griffin et al. [5] pioneered the planar extraction approach, delineating landable planes using concave polygons in a 3D space. This methodology facilitated the vertical projection of 2D footsteps to generate 6D contacts for foot landing, incorporating a provision for approximated 3D collision avoidance. However, it primarily focused on strategies for handling locomotion dynamics centered on foot contacts and did not extend to the intricate requirements of hand contact generation. A notable limitation is the inherent difficulty in utilizing vertical projection for hand contact generation, a critical aspect of multi-contact motion planning, which requires a more comprehensive approach capable of encompassing projections in varied directions, not limited to the vertical.

In noteworthy research, P. Kaiser et al. [2,13] and S. Brossette et al. [14] embraced planar extraction techniques, facilitating collision avoidance through 3D convex body approximations, thereby unlocking the full potential of the GJK algorithm for collision verification. While foundational, this approach necessitates the extensive segmentation of the environment into convex hulls, which can occasionally overly simplify the actual environment and result in inaccurate collision avoidance. Although employing a denser mesh SLAM, such as Kimera [15], can enhance the precision of collision avoidance, it also introduces complexity in identifying polygons proximate to a robotic link.

To resolve the outlined challenges, our research proposes a novel framework that leverages semantic voxels and 3D grid sampling for representing online environments. The inherent Octree structure of our semantic voxels renders detailed 6D contact poses from $\left(x,y,z\right)\in R^{3}$ coordinates, enhancing the scope of 3D grid sampling to encompass all directional planes, not just limited to the vertical direction. This strategy aligns favorably with collision verification protocols, streamlining the generation of contact poses online.

Our system offers a notable advantage by allowing the root link freedom during the planning phase through whole-body optimization. Although recent QP-based controllers for multi-legged robots tend to simultaneously calculate all joint angles, accounting for dynamics, kinematics, and stability, this approach can render offline-planned joint angle configurations unfeasible. To mitigate this effect, we provide only reference trajectories for the end-effectors and allow the QP solver to determine optimal joint angles in real time.

## 3. System Overview

### 3.1. System Architecture

Before delving into the specifics of our method, it is essential to outline the basic prerequisites and provide a succinct introduction to the overall system architecture. Figure 4 provides a concise overview of the architecture of our entire system. The ExOctomap server and our planner operate on an Event-Driven Architecture, continuously monitoring external events such as new point cloud messages, ExOctomap updates, and contact planning requests. Firstly, the robot is equipped with a standard RGB-D camera, specifically the Azure Kinect, mounted on its head. This camera provides a point cloud data stream at a frequency of 30 Hz. Secondly, our ExOctomap system processes this incoming point cloud data to generate a semantically enriched 3D voxel map. Thirdly, each time the robot initiates a swing motion, our replanning system swiftly identifies the next contact point and generates a collision-free swing trajectory to reach it, using the semantic map as a guide. Finally, the updated contact targets are fed into a QP-based controller. This controller employs a QP equation comprising a multi-objective function along with various constraints. Reference CoM trajectories and reference wrenches of the whole robot are computed by Preview Control [16], modeling the humanoid robot as a centroidal dynamics system [17]. The QP controller optimally solves Inverse Kinematics and obtains reference joint angles with a 500 Hz update rate.

### 3.2. QP Formulation

The following mathematical formulation briefly shows our QP controller. To clarify the equations, we define *x* as shown in (Equation 1)
(1)$$\begin{array}{c} x:={\left(q^{T},{\dot{q}}^{T}\right)}^{T} \end{array}$$
(2)$$\begin{array}{cc} \underset{\ddot{q}}{\mathrm{minimize}}\; & \sum_{i=1}^{N}w_{i}{∥{\mathrm{Err}}_{i}\left(x,\ddot{q}\right)∥}^{2} \end{array}$$
(3)$$\begin{array}{cc} \mathrm{subject}\;\mathrm{to}\; & A\left(x\right)\ddot{q}\leq b \end{array}$$
(4)$$\begin{array}{cc} \; & b_{lower}\left(x\right)\leq \ddot{q}dt\leq b_{upper}\left(x\right) \end{array}$$

The target variables for minimization are joint accelerations, $\ddot{q}$. Joint velocities $\dot{q}$ and positions q are obtained through the temporal integration of $\ddot{q}$.

Equation (Equation 2) constitutes the objective function, which is a weighted sum of multiple task errors. For instance, the Center of Mass (CoM) task aims to minimize the discrepancy between the robot’s actual CoM and a reference CoM configuration derived from the online Preview Control [18]. Additionally, the objective function encompasses an orientation task for the base link that guides the link’s orientation toward a desired state. In this study, the planned swing trajectories for the end-effectors are incorporated into a term of the objective function. Each error of tasks is defined as a linear equation of $\ddot{q}$ (Equation 5).
(5)$$\begin{array}{c} {\mathrm{Err}}_{i}\left(x,\ddot{q}\right):=C\left(x\right)\ddot{q}+d\left(x\right) \end{array}$$

Conversely, the QP formulation also incorporates a multitude of inequality constraints. An inequality constraint (Equation 3) contains joints torque limits, calculated by a whole-body inertia matrix M(q), and self-collision margins between links. The other inequality constraint (Equation 4) limits the upper and lower bounds of joint velocities and positions.

As a result, the controller progressively steers the robot’s limbs toward the reference swing trajectory while adhering to the various complex constraints inherent to humanoid robots. As this study concentrates on aspects of robotic vision and motion planning, a more exhaustive discourse on the subject can be found in the original paper [19]. It is worth noting that similar QP formulations have also been employed for quadruped robots, as seen in the work by C. Mastalli et al. [20] and M. Risiglione et al. [21].

## 4. Vision (ExOctomap)

We have improved the existing Octomap [4] approach, thereby enhancing its feasibility for humanoid robots interacting with their surroundings without the need for static 3D models. Our upgraded version, termed ExOctomap, is designed to enhance the speed of motion planners, owing to the rich data stored in each of its voxels.

The ExOctomap is formed of thousands of small units called voxels. As shown in Figure 3, each voxel stores important information, as follows:OccupancyNormal vector $\vec{N}\in R^{3}$Object primitive class (e.g., stair, floor, wall, ceiling...)Object’s PCA parameters (eigenvectors $\vec{e_{i}}\in R^{3}$ and corresponding eigenvalues $\lambda_{i}\left(i=1,2,3\right)$)Color (optional. only used for visualization)

This information is organized within an Octree structure, set in a world coordinate system. This approach allows our planner to rapidly determine the details stored at any point ${\left(x,y,z\right)}_{world}$ with an efficiency of *O*(log *N*). By incorporating semantic segmentation results within each voxel, planners can now swiftly access vital data such as the normal vector and PCA results by accessing the relevant voxel.

Algorithm 1 briefly explains the process flow of our ExOctomap. The ExOctomap method involves two main steps: (1) map generation and (2) semantic segmentation of all data.
**Algorithm 1** Update of ExOctomap (REVISED)**Input:** *current map*, *point cloud* and *camera pose* in world coordinate**Output:** *updated map* 1:*point cloud* ← *camera stream* 2:voxelize(pointcloud) // twice resolution of ExOctomap 3:calc $\vec{N}\left(point\;cloud\right)$ // normal vector estimation 4:transform(pointcloud,camerapose) 5:WindowCropping(pointcloud,camerapose) 6:**for** each point in *point cloud* **do** 7:   $voxel_{point}\leftarrow access\left(pos_{point},current\;map\right)$ 8:   **if** $voxel_{point}$ is NULL **then** 9:     $allocate(voxel_{point},current\;map)$10:   **end if**11:   $update(voxel_{point},\vec{N})$12:**end for**13:WindowCropping(currentmap,camerapose)14:clusterslist←ECE(currentmap)15:**for** each cluster in clusterslist **do**16:   ${\vec{e}}_{1,2,3},\lambda_{1,2,3}\leftarrow PCA\left(cluster\right)$17:   $label\leftarrow classify(cluster,{\vec{e}}_{1,2,3},\lambda_{1,2,3})$18:   $voxel_{cluster}\leftarrow access\left(point_{cluster},current\;map\right)$19:   $update(voxels_{cluster},label,{\vec{e}}_{1,2,3},\lambda_{1,2,3})$20:**end for**21:RayCast(currentmap,camerapose)22:updatedmap←currentmap23:broadcast(updatedmap) // motion planner will receive it via ROS communication

Note that our ExOctomap functions solely as a mapping system, similar to the original Octomap, and is not a 3D-SLAM system that estimates camera poses in the world coordinate system. In this research, the robot updates its self-localization using only IMU sensor readings and joint encoder values rather than vision-based methods. We explored this particular approach in a previous publication [22].

### 4.1. Voxel Generation

In Algorithm 1, lines 2 to 12 correspond to the processes detailed in this section.

The inputs required are a stream of point cloud data along with the respective camera pose in 3D space. Similar to the original Octomap, a transformation matrix facilitating the conversion from world coordinates to the camera frame is essential for 3D mapping.

The process unfolds as follows:Implement a twice resolution of voxelization filter on the incoming point cloud data.Determine the normal vector for each individual point.Transition the point cloud data to the world coordinate system.Create a voxel and allocate the normal vector to it, or update the vector if the voxel pre-exists.

The initial step of applying a voxelization filter serves to expedite subsequent processes; however, this approach compromises finer details. To maintain computational precision for normal vector, the resolution during this voxelization process is set to be twice that of the ExOctree itself.

The transformation of the point cloud is facilitated through the ROS’s TF (Transformation) [23] framework. This framework queries the most recent cached data for the camera pose within the global coordinate system, subsequently utilizing the corresponding transformation matrix that aligns temporally with the point cloud observation. Although signal latency can pose a significant challenge for systems requiring high-speed motion, our robotic system operates at a relatively conservative speed. Moreover, our methodology adopts a voxel-based representation scheme, akin to the original OctoMap, to manage the geometric complexities of the environment in a probabilistic manner. This approach dynamically allocates and deallocates voxels by consistently integrating incoming point cloud data streams. Consequently, the impact of latency on perception accuracy was deemed negligible in the context of this research.

### 4.2. Window Cropping

In the Algorithm 1, two distinct window cropping processes are invoked at lines 5 and 13, acting upon the input point cloud and the comprehensive map, respectively. These cropping procedures serve dual purposes: first, to conserve the memory allocation for the map, and second, to expedite the semantic segmentation process. Through empirical evaluation, the dimensions of the cropped area were set to a 3.0 m × 3.0 m grid centered at the camera’s current location.

The mapping region is dynamically adjusted to align with the robot’s spatial trajectory, thereby preserving the fidelity of both tracking and data collection mechanisms. The initial cropping phase at line 5 specifically eliminates distantly-located points, which generally bear minimal relevance to our contact planning algorithm. The subsequent cropping operation at line 13 purges antiquated voxels positioned behind the robot.

As a result, our ExOctomap maintains a dynamic and relocatable presence, emulating the movements of the robot within its environment.

### 4.3. Semantic Scene Segmentation

In Algorithm 1, lines 14 to 20 pertain to scene recognition and voxel update processes. Figure 5 provides a visual depiction of the subsequent processes.

During the voxel generation stage, every voxel inherently incorporates a normal vector. Subsequently, we aggregate all voxels in the ExOctomap into distinct clusters through the application of the Euclidean Cluster Extraction (ECE) algorithm [26], leveraging both their spatial proximity and associated normal vectors. This facilitates differentiation between structures with varying orientations, thereby delineating walls into separate planes and segregating staircases into horizontal and vertical segments.

Prior to the clustering executed via the ECE method, each cluster is analyzed and categorized based on PCA. This analytical process discerns the eigenvectors and eigenvalues intrinsic to each voxel assembly, thereby offering insight into voxel distribution and facilitating the recognition of the shapes represented by different clusters. In this study, we have predicated the classification of object primitives on their individual sets of eigenvectors and eigenvalues.

Our methodology is adept at pinpointing three primitive shapes: cylinders, planes, and spheres. When examining cylindrical entities like handrails or poles, the principal eigenvector indicates the direction of the object, a feature instrumental in ascertaining the yaw angle for hand-grasping postures. Furthermore, planes are differentiated into vertical or horizontal categories based on the orientations of their third eigenvectors, which correspond to the normal vectors, thus enabling the planner to swiftly decide the feasibility of foot placement.

The clusters are identified as spherical objects when the eigenvalues exhibit close magnitudes. During experimental observations, such spherical clusters predominantly manifested in human subjects, attributed to the continuous alterations in their normal vectors.

### 4.4. Update Semantics with Memory Queue

This section corresponds to line 19 in Algorithm 1. Traditional point cloud methodologies such as normal vector estimation, ECE, and PCA generally encounter challenges concerning stability when handling one-shot point cloud data. To enhance robustness and introduce a time dimension, we instituted a majority voting system that takes into account the historical semantic analysis data for each voxel.

Due to various disturbances, including camera noise, random fluctuations in sampling, and inaccuracies in normal vector estimation, the ECE clustering and PCA processes occasionally yield incorrect detections. Drawing inspiration from the original Octomap’s feature of gradually updating a voxel’s occupancy status based on a series of sensor inputs over time, our ExOctomap implements a majority voting system in the realm of semantic segmentation, fostering more reliable and accurate mapping.

In this system, each voxel maintains a queue to archive a series of recent results, thereby facilitating a dynamic yet orderly update process. When allocating a new semantic label to a voxel, it discards the oldest data, integrates the new input into the queue, and deploys a majority vote to establish the final label (Figure 6). In our trials of Section 6, we adopted a memory queue length of 10.

### 4.5. Ray Casting for Dynamic Obstacle Removal

In Algorithm 1, line 21 corresponds to the procedures described in this section. To facilitate the robot’s adaptability to real-time environmental alterations, we refined the ray-casting functionality inherent in the original Octomap [4]. Our ExOctomap exhibits an enhanced capacity to smoothly dispose of outdated voxels compared to its predecessor.

Figure 7 illustrates that the initial version of the ray-casting algorithm adopts a conservative approach, establishing rays from the camera to individual points within the input point cloud data. This strategy is limited to deleting voxels that are directly aligned with these rays. Although this approach is beneficial as it reduces computational demand with the use of fewer rays, it has the drawback of delaying the removal of obstacles.

In scenarios encompassing expansive environments or receiving sparse input from point cloud data, the system fails to eradicate voxels originating from temporary events, such as a person crossing the camera’s field of vision. This failure is attributed to the absence of rays targeting these vacant areas. This limitation was clearly observed in our real-world tests, where remaining voxels often disrupted the creation of paths.

To rectify this error, our ExOctomap deploys a considerable array of rays within the camera’s field of observation, thereby diminishing voxel occupancy. Although this adjustment presently induces a lag in the loop latency of the mapping system, it guarantees a more uncluttered map. During the robotic trials, the swing phases consistently exceeded a duration of 1.5 s, ensuring that prior to initiating the subsequent motion planning sequence, the planner receives this refined map within adequate time (Figure 8).

### 4.6. Comparison with Original Octomap

To quantitatively evaluate the efficiency of ExOctomap compared to the original Octomap [4], a benchmark was established. A point cloud stream and corresponding camera poses were recorded while traversing a distance of approximately 7.0 m. Both the original Octomap and ExOctomap were then fed this data for mapping, with voxel resolutions uniformly set to 3.0 cm. It is important to note that the timing of data sampling varied due to the immediate commencement of subsequent loops upon the completion of previous ones.

Figure 9 presents the processing speed for each mapping system, defined as the duration from the receipt of new point cloud data to the output of an encoded communication message. Despite the additional computational requirements of semantic segmentation and ray-casting, ExOctomap exhibited a faster execution time compared to the original Octomap. This suggests that the initial voxelization process, as implemented at line 2 in Algorithm 1, effectively reduced computational overhead in subsequent steps, specifically in normal vector estimation and transformation to the world coordinate system.

To further validate this observation, voxelization was manually disabled for subsequent comparisons. Figure 10 illustrates that under this condition, ExOctomap was unable to maintain a stable processing time. Voxelization is commonly employed in point cloud processing to enhance computational efficiency by reducing data point counts while preserving uniform point cloud density. Although there is a potential trade-off in shape feature precision, the map remained accurate for the robot to interact with the environment. All subsequent experiments reinstated the voxelization filter.

Figure 11 illustrates the number of voxels managed by each mapping server. Unlike the original Octomap, which experiences fluctuations in voxel count due to its conservative ray-casting function, our ExOctomap maintains a stable number of voxels. This stability is particularly crucial for the semantic segmentation process, which operates on the entire map. By confining the generation and retention of voxels to a predefined area through its window cropping function, ExOctomap not only ensures consistent processing speed but also enhances the efficiency of semantic segmentation.

In terms of memory requirements, each voxel in the original Octomap occupies 16 bytes, whereas ExOctomap’s voxels require 248 bytes owing to additional semantic information. Consequently, while the original Octomap utilized a total memory of 11.87 Mega Bytes to manage up to 774,461 voxels, ExOctomap consumed 14.12 Mega Bytes for a substantially smaller voxel count of 59,714. Figure 12 presents the memory consumption for each map throughout the benchmark test. Although the shape of the orange line mirrors that in Figure 11, it appears more precipitous due to scaling factors.

## 5. Motion Replanning

The preceding sections focused on vision recognition techniques. Hereafter, we elucidate the motion planning component. This is detailed in Algorithm 2, which delineates our approach to contact replanning. Leveraging the semantics contained within the ExOctomap voxels, our robotic system dynamically adapts the contact points according to the actual environment online.

In the intervals between swing phases, the planner retrieves the current state of the robot to promptly devise the subsequent transition motion. Note that the present study does not engage in modifying the sequence of contact transitions; deciding whether a subsequent hand contact is necessary or not will be considered a research topic for future work.

Our replanning process consists of

(a).Grid sampling around the original contact position(b).Sort by modification cost (L2 norm)(c).Local collision check of a contact candidate(d).Generate swing trajectory with spline(e).Trajectory collision check

**Algorithm 2** Contact replanning algorithm
**Input:** initial 6D contact $c_{init}$, robot state robot, ExOctomap**Output:** modified 6D contact $c_{modif}$ and its collision-free trajectory *traj*
 1:

$voxel_{pivot}\leftarrow discretize\left(c_{initial}\right)$

 2:

$voxels_{available}\leftarrow GridSampling\left(voxel_{pivot},primitive,l_{sampling}\right)$

 3:

$sort(voxels_{available},c_{init},L2)$

 4:**for** $voxel_{cand}$ in $voxels_{available}$ **do** 5:   $c_{cand}\leftarrow generate\left(voxel_{cand}\right)$ 6:   $LocalCollisionCheck(voxels_{EndEff},c_{cand},ExOctomap)$ 7:   **if** $c_{cand}$ is locally collision-free **then** 8:     $list_{feasible}\leftarrow c_{cand}$ 9:   **end if**10:
**end for**
11:**for** $c_{cand}$ in $list_{feasible}$ **do**12:   $CoM_{ref}\leftarrow WeightedAverage\left(c_{cand},robot\right)$13:   $scfr\leftarrow DrawSCFR(c_{cand},robot)$14:   **if** $isProjectable(CoM_{ref},scfr)$ **then**15:     $traj\leftarrow GenerateSwingTraj(c_{cand},robot,ExOctomap)$16:     **if** a collision-free trajectory found **then**17:        $c_{modif}\leftarrow c_{cand}$18:        break19:     **end if**20:   **end if**21:
**end for**
22:**return** 
$c_{modif},traj$


In our method, the initial step involves generating a substantial number of contact candidates, a process executed through two primary stages: (a) grid sampling to develop an array of potential contact points, followed by (b) arranging these candidates in ascending order according to their modification costs, assessed through the L2 norm in a 6D space. This organization facilitates a prioritized evaluation of each candidate based on the presumed efficiency in achieving contact.

With the candidates duly sorted, we proceed to (c) assess the feasibility of each through a meticulous local collision check. The planning algorithm subsequently (d) crafts a reference swing trajectory, leveraging cubic spline interpolation to maintain a smooth transition between waypoints. Moreover, to ascertain the viability of the crafted trajectory, (e) a collision check is performed against the environmental voxels represented in the ExOctomap voxels.

When a collision is detected at stage (e), the system reinitiates the trajectory generation at point (d), altering the waypoints to formulate a different approach path. This iterative process continues until a candidate showcasing a collision-free trajectory is identified, signaling the successful conclusion of the replanning phase. This expeditious process yields a two-part output: a designated 6D contact target paired with a refined reference swing trajectory.

We implemented this algorithm using C++ by fully leveraging multi-threading to enhance the performance efficiency. Remarkably, the entire sequence, spanning stages (a) through (e), is executed in approximately 50 ms, demonstrating that the method is feasible for real-time applications.

### 5.1. Grid Sampling and Local Collision Check

This section is relevant to lines 2 and 6 in Algorithm 2. By utilizing the semantic information contained within the environmental voxels, our planner efficiently generates a series of 6D contact pose samples while reducing the dimensionality of the search space from 6D to either 3D or 4D.

Figure 13 showcases the outcome of utilizing grid sampling for the generation of contact candidates. By assuming manually defined contact targets or cyclic patterns as a reference motion, our system develops 6D contact candidate poses. This is achieved by uniformly allocating grid points in the vicinity of the initial contact positions within a cubic sampling region defined by a specified side length, $l_{sampling}$. The semantics allow the planner to promptly identify the feasible voxels for each end-effector and contact type.

To validate the semantics of each voxel, the grid sampling operates with the same resolution as the ExOctomap, which is precisely 3.0 cm in all experiments in Section 6. The normal vector in sampled voxels allows for efficient determination of grasping or landing orientations with *O*(log *N*) efficiency. These normal vectors represent local features of the voxel, in contrast to the global features depicted through the results obtained from ECE and PCA.

In cases of grasping handrails, the roll and pitch angles are aligned with the normal vector, whereas the yaw angle conforms to the eigenvector inherent to the voxel. Conversely, actions such as feet landing or a palm pushing against a flat plane grant one degree of freedom in the yaw axis, thereby yielding multiple candidate orientations generated at defined intervals.

Following the generation of the contact poses grid, the planner scrutinizes the local collision and stability of each configuration, retaining only those configurations that are validated as feasible contacts.

Local collision checkFigure 14 illustrates the local collision checking function that is integral to our approach. Traditionally, collision detection has been simplified through the utilization of a collision sphere set, a method that tends to preclude possibilities, especially when the end-effector is in close proximity to the object of contact [27]. To address this limitation, we adopt a strategy wherein a voxelized representation of the end-effectors is engaged, thereby enabling the precise evaluation of local collisions occurring between the voxels of the end-effector and those constituting the environment.Notably, distinct from the trajectory collision check delineated in Section 5.2, this process is strictly localized. Following this pathway ensures the elimination of contact poses that engender collisions with the environmental voxels, enhancing the accuracy and safety of robot operations.Concurrently, the planner affirms the number of voxels of the end-effector that are in contact with the environmental constituents, which is a pivotal step in ascertaining a sufficiently expansive contact area. This is a prerequisite for mitigating the robot’s oscillatory tendencies and affirming its stability. The process is guided by a predetermined minimum threshold for contact ratio, beyond which contacts are deemed insufficient and are consequently disregarded.Static static stability checkThis subsection delineates the procedures articulated in lines 12 through 14 of Algorithm 2. The stability of the subsequent configuration is assessed utilizing the Static CoM Feasible Region (SCFR) [3,28]. Initially, the reference CoM position is computed through a weighted average of the current contact points combined with each target contact candidate. Subsequently, by projecting the existing location of the CoM onto the SCFR polygon, the planner is enabled to discern and dismiss candidates that are statically unstable.

After performing the aforementioned collision and stability assessments, only the contact candidates that are locally feasible are retained. Given that the subsequent trajectory generation process demands a higher computational overhead, our method prioritizes the elimination of unviable contacts via these preliminary, less computationally intensive checks.

Although several studies have focused on determining contacts based on long-term transition feasibility, our research prioritizes minimizing changes to the contact point, an approach that considers the unpredictable nature of unknown environments. This strategy helps prevent the robot from undergoing large deviations from its initial planned motion. Subsequently, the planner organizes all candidates in an ascending order based on the L2 norm in the 6D space. Starting with the candidate reflecting the most minimal modification, the system strives to delineate a collision-free trajectory, originating from the limb’s current position to the subsequent contacts.

### 5.2. Trajectory Generation with Collision Check

This section elucidates the *GenerateSwingTraj* function detailed in Algorithm 2, with Figure 15 visualizing our trajectory generation procedure based on cubic spline interpolation.

In tasks involving walking or multi-contact motions, which fundamentally entail a repeated cycle of engaging and disengaging with the environment, the requisite precision is generally less compared to object manipulation tasks. Based on this premise, we conceptualize the swing motion as a curve, drawing a reference limb trajectory utilizing cubic spline interpolation.

Mirroring the approach adopted for local collision checks delineated in the preceding section, we employ a voxelized model. N. Perrin et al. [29] leveraged 3D models of obstacles, markers, and external cameras for obstacle recognition in their proposition for collision checking with a robot’s foot motion’s swept volume, whereas our research operates within the constraints of unknown environments, precluding the utilization of 3D models, markers, or external cameras. Consequently, we resort to the direct application of voxels as collision models for environments, a strategy similarly executed by A. Hermann et al. [30,31].

Initially, we establish two waypoints: a waypoint above the current contact point representing the starting position and another waypoint above the designated target contact point denoting the goal position. Subsequently, the planner draws a cubic spline curve and examines the trajectory at discretized intervals to identify any environmental collisions.

When a collision is detected on the front side of the end-effector, the strategy serves to elevate the waypoint of the starting side, thereby avoiding the obstacle. In contrast, when collisions occur on the back side of the end-effector, the response involves raising the waypoint of the goal side. Notably, the latter modification frequently plays a pivotal role during downward climbing motions.

Due to the substantial computational resources required for collision checking over the trajectory, our planner does not exhaustively search through all contact candidates. In particular, it is designed to halt immediately upon determining a collision-free trajectory for a given contact candidate, effectively concluding the replanning process at this juncture. This newly devised motion plan is promptly incorporated into the QP objective function. Subsequently, the QP solver determines the optimal joint angles that would realize the end-effector’s motion, thereby realizing this task within a timeframe of 2 ms.

## 6. Experiments

We used the exact same parameters for the following simulation experiments and real experiments, except for the reference contact patterns.

Leg swing duration: 1.5 sHand swing duration: 7.0 sVoxel map resolution: 3.0 cmWindow cropping size: 3.0 m × 3.0 mContact searching area: x: 48.0 cm, y: 30.0 cm, z: 166.0 cmDisplacement of waypoints for trajectory planning: 3.0 cm

### 6.1. Simulation

We conducted system tests within the dynamic simulator Choreonoid, utilizing a virtual stream of point cloud data. Initially, the HRP5P robot was equipped with a contact sequence not calibrated for actual environments, necessitating adjustments to its motion based on vision-sensing outcomes. In response, our replanning system created contact points and transition trajectories, which were integrated into a Centroidal model-based QP controller. This QP controller was responsible for determining all necessary joint angles to facilitate end-effector tracking as the robot moves, ensuring stability and adherence to additional constraints while maintaining dynamic balance throughout its operation. The details of the QP controller are in [18].

In the simulation experiments, we utilized a computer equipped with an Intel Core i9 processor (3.6 GHz, 8 cores), 32 GB of RAM, and an NVIDIA GeForce RTX2080(Super) GPU. This system was tasked with managing the dynamic simulator alongside real-time virtual point cloud generation, execution of the robot’s QP controller, operation of our planning algorithm, and running our ExOctomap system.

#### 6.1.1. Scenario 1: Surmounting a Hurdle

Figure 16 illustrates HRP5P navigating over a minor obstacle. Initially, the motion data stipulated flat, bipedal walking with a stride length of 25.0 cm. Regardless, the vision and replanning system discerned an obstacle in line with the original plan. To counteract this obstacle, it autonomously crafts swing trajectories that situate the landing point either shortly before the obstacle or facilitate a detour around it. The system invoked the replanning mechanism over a total of 17 iterations, with operations lasting an average of 24.5 ms; the durations spanned between 13.8 ms and 90.8 ms. The most time-consuming task involved formulating the collision evasion motion for the left foot, a process illustrated in the second image of Figure 16. Notably, if the mapping system solely extracts available planes, it overlooks small obstacles, thereby underscoring the importance of preserving the original contour of the environment for successful collision avoidance. Although the grid sampling technique occasionally yields contact candidates in proximity to the obstacle, these are systematically eliminated in the local collision checking or trajectory generation stages, thereby maintaining a collision-free motion for the robot.

#### 6.1.2. Scenario 2: Grasping Handrails

Figure 17 delineates a scenario of handrail grasping, elucidating the robustness of our planning approach in adapting to erroneous initial conditions characterized by substantial discrepancies in the grasping contacts, which range from 12.6 cm to 20.8 cm. Despite the foundational motion data being premised on flat bipedal walking with 25.0-cm strides, our planner adeptly performed recalibration to design a 3D swing trajectory for the footsteps while precisely aligning the hand contacts with the handrails. A computation was executed within an average timeframe of 48.3 ms, with a variance ranging from 16.8 ms to 74.1 ms.

A critical analysis of individual frames revealed that the longest processing span was allocated to the right foot’s pre-transitional phase near a diminutive triangular plate, evident in the terminal imagery of Figure 17. This extended duration, peaking at 74.1 ms, was necessitated due to the intricate task of pinpointing a partial contact on the irregular surface, a venture that was ultimately forgone as the robot landed just before the triangular plate. Conversely, a more streamlined adjustment is observed in the second snapshot of Figure 17, where a short period of 27.5 ms facilitated a 12.6 cm rectification in contact positioning, fostering a trajectory devoid of collisions.

The planner’s aptitude in swiftly recognizing 6D grasping contenders from the 3D grid sampling is leveraged through the intravoxel storage of global directional data pertinent to the pipes, enhancing the efficiency of the identification process. However, it encounters a limitation in the fourth frame, hindered by the insufficient contact space on the triangular plate, which failed to meet the predefined minimum contact ratio for the right foot to navigate the descending spiral staircase. This poses a considerable challenge in reducing the threshold without compromising the stability and introducing a precarious balancing act dictated by a minuscule contact area. Although this ratio can be reduced in the planner settings, it entails another problem: maintaining the balance of the robot with a smaller contact area. To address this challenge, the contact size must be dynamically changed to manage the pressure more finely across a smaller surface.

### 6.2. Real World

In this study, we conducted experiments in the real world. We employed a dual-computer architecture. The computer previously used for simulation experiments, equipped with an Intel Core i9 processor, 32 GB of RAM, and an NVIDIA GeForce RTX2080 Super GPU, was repurposed for the generation of the ExOctomap. The second computer of Intel NUC, integrated within the robot, was designated for motion planning and control tasks. The externally generated ExOctomap was transferred to the robot’s onboard system via ROS communication. Upon receipt, the motion planner decodes the incoming data, reconstructs the Octree structure, initiates our proposed replanning algorithm, and passes the resultant motions to the QP controller.

#### 6.2.1. Scenario 1: Surmounting Blocks

Figure 18 illustrates the HRP5P successfully navigating a pathway obstructed by concrete blocks. Initially endowed with a reference motion predicated on a cyclic bipedal walking pattern with 25.0-cm steps, the robot was expected to dynamically adapt to the changing environment. During the active phase of the system, an individual placed blocks in the robot’s path, thereby altering the previously flat environment and generating additional voxels from the person’s movements. Regardless, the ray-casting function effectively removed the unnecessary voxels, which enabled our planner to delineate suitable collision-free trajectories for the robot limbs. Throughout the experiment, the replanning function was triggered for 12 instances, with the computational time averaging at 23.5 ms, varying from 19.2 ms to 27.3 ms. Notably, the segment requiring the most extensive computation time involved strategizing the left foot’s elevation to surmount a block.

#### 6.2.2. Scenario 2: Walking with Table Supports

Figure 19 illustrates a scenario wherein the HRP5P robot maneuvers while establishing contact with tables. Initially, the robot features a pre-defined cyclical contact sequence, which alternates between the left foot, right foot, and left hand, each advancing by 25.0 cm, serving as the reference for its motion. Regardless, the real-world environment introduces unforeseen obstacles, including concrete blocks and a small hurdle, that are not considered in the initial reference contacts.

Our vision and planning system adeptly recognized viable contact points online, along with detecting potential collision hindrances. This approach allowed for the creation of new contact points and collision-averse transition trajectories at every step, enhancing the robot’s navigational abilities. Throughout the operation, the replanning function was triggered 22 times, with an average processing time of 48.6 ms, fluctuating from 38.3 ms to 115.7 ms.

Addressing the obstacle posed by the small hurdle, specifically modifying the left hand’s movement, necessitated a mere 47 ms of computational time by our planner, as depicted in the top-middle and top-right images of Figure 19. Conversely, orchestrating the right leg’s stepping motion between two concrete blocks demanded maximum computational resources. This complexity arose as the planner persistently formulated a landing strategy on the second block, a task complicated by the absence of collision-free trajectories on the targeted surface. The video is available at https://youtu.be/a1Q6IB2DRes (accessed on 27 September 2023).

## 7. Discussion and Future Works

In the prevailing framework of our system, the trajectory delineated by the planner serves as a reference for the end-effector within the objective function of the QP-based whole-body controller, as stipulated in [18]. Throughout the conducted experiments, which featured environments characterized by their extensive and sparse nature, there were no recorded instances of the robot’s elbows or knees encountering collisions with obstacles present in the environment. Nevertheless, the existing system framework does not inherently guarantee against potential collisions of this nature. To rectify this issue, real-time inequality constraints must be incorporated to facilitate the maintenance of a safe distance from the proximate voxel. Current research focuses on devising a rapid search algorithm equipped to search for the nearest voxel from the whole map, a feature intended to foster the introduction of a novel inequality constraint to the QP controller.

As another constraint of our system, our replanning strategy is currently designed to initiate the next contact planning process only after landing. Previously, we explored the possibility of executing the planning phase concurrently with the swing motions, aiming to craft the next swing trajectory while an end-effector was still airborne. This approach encountered challenges, as the QP occasionally failed to converge to the prescribed motion, leading to slightly misaligned contacts. Moreover, landing could induce a shock reaction in the robot, prompting it to adjust its foot contacts to absorb the impact, resulting in a minor alteration in the robot’s position upon landing. This discrepancy between the pre-constructed swing trajectory and the robot’s actual status post-landing highlighted the necessity for a mechanism allowing real-time trajectory re-adjustments, drawing from the trajectory established during the airborne state.

## 8. Conclusions

In this study, we presented an advanced system capable of guiding multi-contact motions in unknown environments. Many previous approaches based on graph search and optimization have often fallen short in real-world applications. In contrast, our method introduced the concept that semantic voxels can supply motion planners with feasible 6D contact poses and collision-free trajectories. Our ExOctomap effectively manages semantic information within voxels and Octree structure. This semantic richness enables efficient and rapid 3D sampling for generating precise 6D contact poses. Utilizing 3D vision, rapid planning, and QP-based whole-body control, the system operates at a remarkable computational speed, making it well-suited for real-world applications. Experimental results confirm the system’s ability to navigate dynamic environments responsively and accurately.

However, the system has limitations, most notably a lack of guaranteed collision prevention near the robot’s elbows and knees. Immediate plans include algorithmic improvements to address these issues. We are also exploring new inequality constraints in the QP controller to augment safety measures.

In summary, this research lays the groundwork for further advancements in robotic adaptability and responsiveness, particularly in unknown environments. Future work will continue to focus on the synergistic integration of 3D vision and control strategies to enhance the robot’s autonomous capabilities.

## Figures and Tables

**Figure 1 sensors-23-08837-f001:**
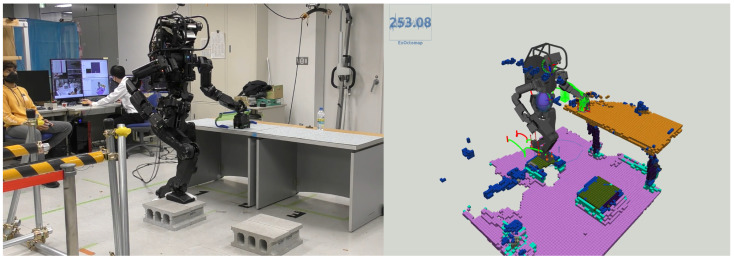
HRP5P is maneuvering over blocks while sustaining contact with the surface of the table. **Left**: The actual environment. Initially programmed with a flat bipedal footstep pattern coupled with cyclic left-hand contact, our method dynamically adapts the robot’s motion to the actual environment. **Right**: Our Extended-Octomap system. The blue number on its left-top (253.08 ms) represents a process time of the mapping system at this moment. Details of this experiment are elaborated in Section 6.

**Figure 5 sensors-23-08837-f005:**
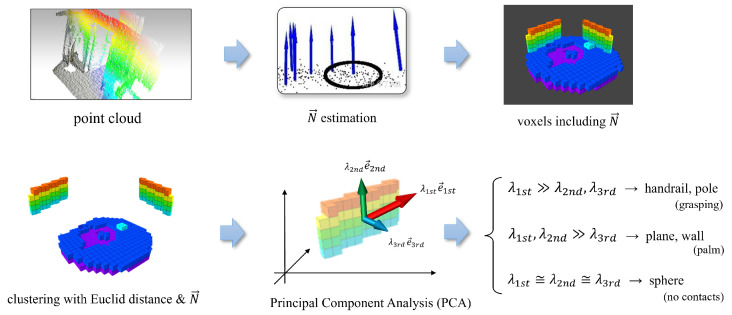
Our instance segmentation process. 1. Division is performed, yielding specific clusters by Euclidean distance and normals. 2. Principal component analysis (PCA) is performed to examine the distribution trend of voxels within a cluster. 3. The PCA results are embedded into all voxels in the cluster (images are partially cited from [24,25] and modified).

**Figure 6 sensors-23-08837-f006:**
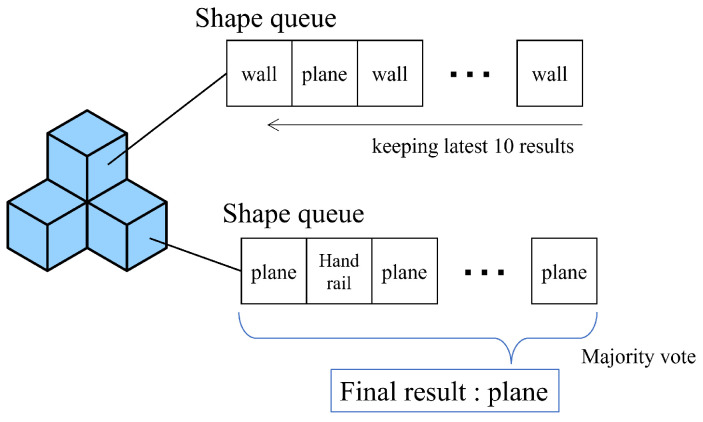
Update strategy for a voxel label, with a majority vote determining the result for planning purposes. We define four categories of shape labels: horizontal planes (such as floors or tables), walls, cylinders, and a cubic category involving other shapes.

**Figure 7 sensors-23-08837-f007:**
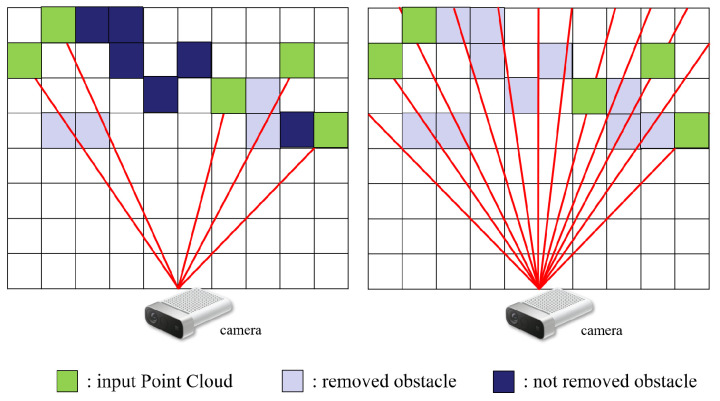
Comparative illustration of ray-casting implementations between the original Octomap and our ExOctomap. **Left**: The original Octomap directs rays exclusively to new input points (highlighted in green). In a 3D environment, these rays seldom intersect with previously established obstacle voxels, thereby hindering the timely deletion of outdated information. **Right**: In contrast, our enhanced version systematically deploys approximately 2500 rays to facilitate a smoother and more efficient removal of old voxels.

**Figure 8 sensors-23-08837-f008:**
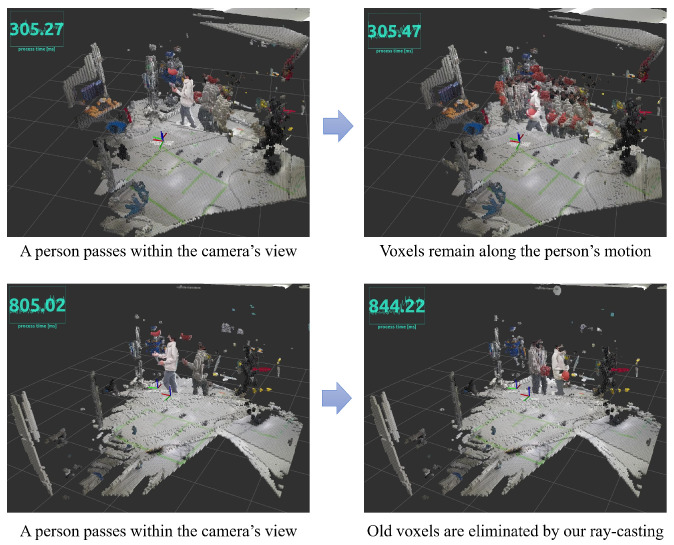
A comparison of old voxel elimination using two different ray-casting methods. Blue numbers represent the whole process time of Algorithm 1 at each moment. (**Upper**) In the original Octomap implementation, ray-casting is conducted only from the camera to the floor, resulting in the removal of only the voxels representing the person’s legs, leaving the torso voxels intact. (**Lower**) In contrast, our ExOctomap implementation performs ray-casting across the full extent of the camera’s field of view, effectively removing all dynamic obstacles. Notably, while this approach is more comprehensive, it requires a longer processing time, averaging 850 ms for each loop in this demonstration.

**Figure 9 sensors-23-08837-f009:**
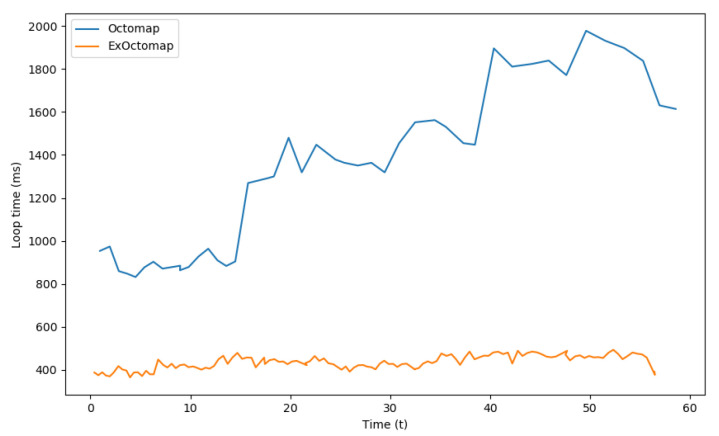
Comparison of loop time between the original Octomap and our ExOctomap. The blue line indicates the process time of the original Octomap taken for each loop. As it takes a longer time, the original Octomap runs only 45 times in 60 s. On the contrary, because our ExOctomap runs much faster, it runs 127 times per loop, even including the semantic segmentation process.

**Figure 10 sensors-23-08837-f010:**
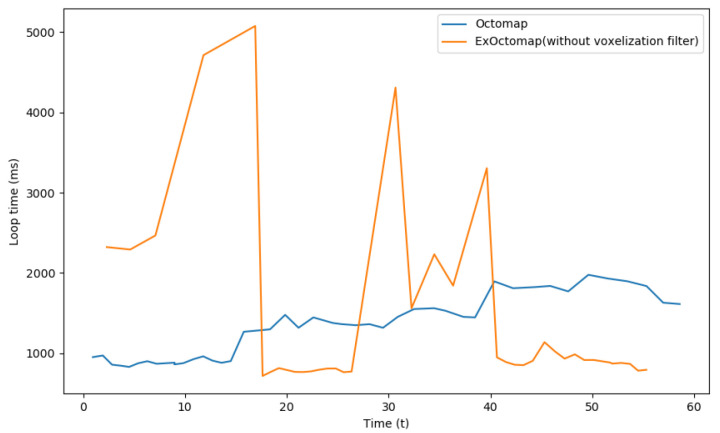
Comparison of loop time between the original Octomap and our ExOctomap without its voxelization filtering process. If our ExOctomap does not apply a voxelization filter to the input point cloud data, the number of rays in our dense ray-casting function incredibly increases. Therefore, process time cannot be stable.

**Figure 11 sensors-23-08837-f011:**
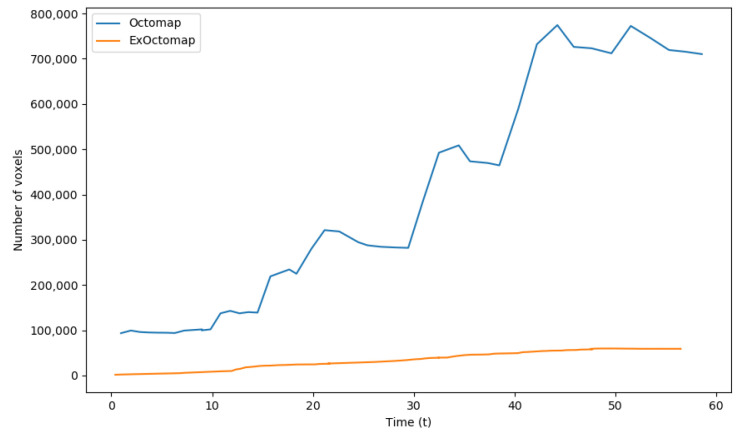
Comparison of voxel counts. These data contain not only the leaves but also the parent nodes of the Octree structure. Although the original Octomap expands its mapping region as the camera advances, ExOctomap utilizes a window cropping function to constrain map size. This function selectively removes distant voxels from the camera’s perspective, thereby maintaining a reduced voxel count.

**Figure 12 sensors-23-08837-f012:**
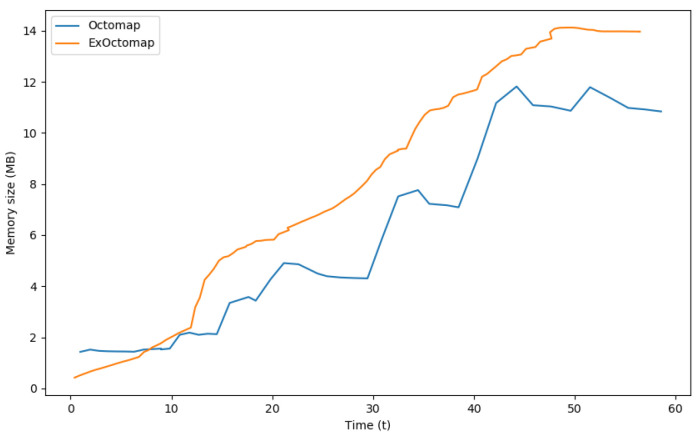
Comparison of memory consumption. Our ExOctomap occupies a larger amount of memory than the baseline method. Because the Octree structure recursively divides a 3D space into 8 sub-cubes, both terminal leaves and their parent nodes occupy the same memory size of 248 bytes. Potential code optimizations, such as dynamic allocation of semantics solely to leaf nodes, could enhance memory efficiency.

**Figure 13 sensors-23-08837-f013:**
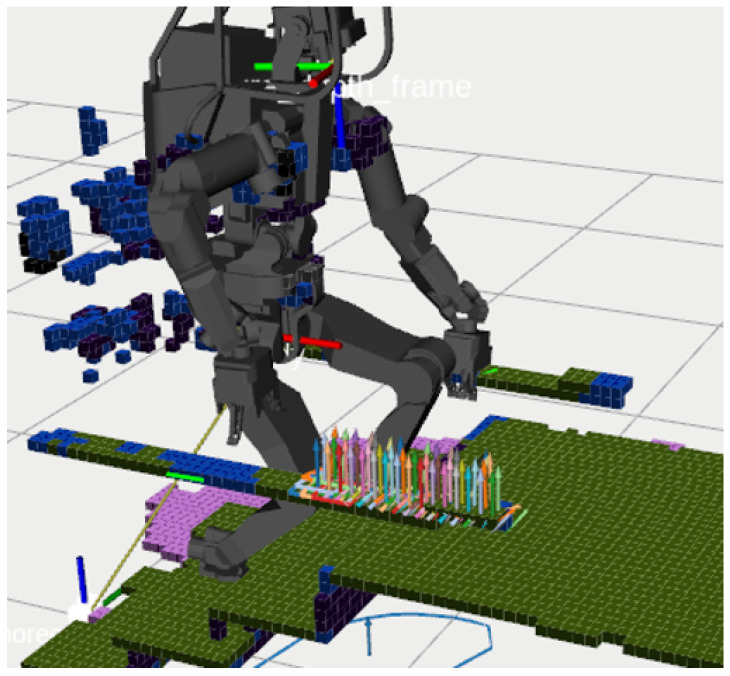
Visualized contact candidates generated through grid sampling, each aligned with the eigenvector of its respective voxel to represent the global direction of the handrail.

**Figure 14 sensors-23-08837-f014:**
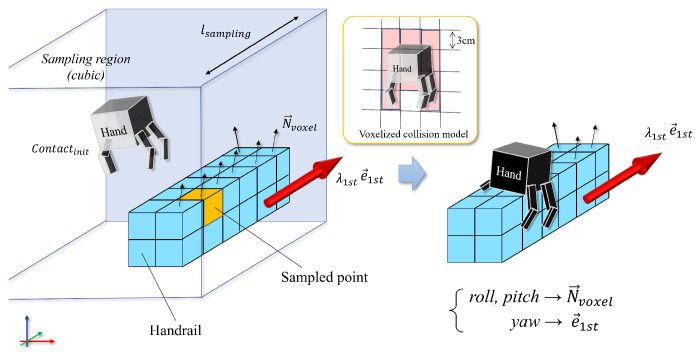
Scene depicting the grid sampling process used for contact generation. In the vicinity of the original contact point, our planner investigates filled voxels labeled with available primitives. Subsequently, it virtually transitions a voxelized collision model to the sampled candidate position to ascertain whether the resulting contact will be collision-free.

**Figure 15 sensors-23-08837-f015:**
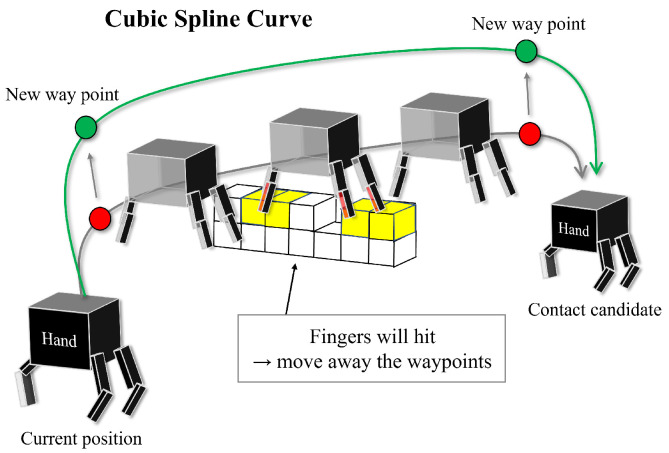
An outline of the collision-free trajectory generation process. The planner constructs a cubic spline trajectory connecting the current position to the designated contact point, iteratively adjusting the waypoints based on the outcomes of the voxel collision checking.

**Figure 16 sensors-23-08837-f016:**
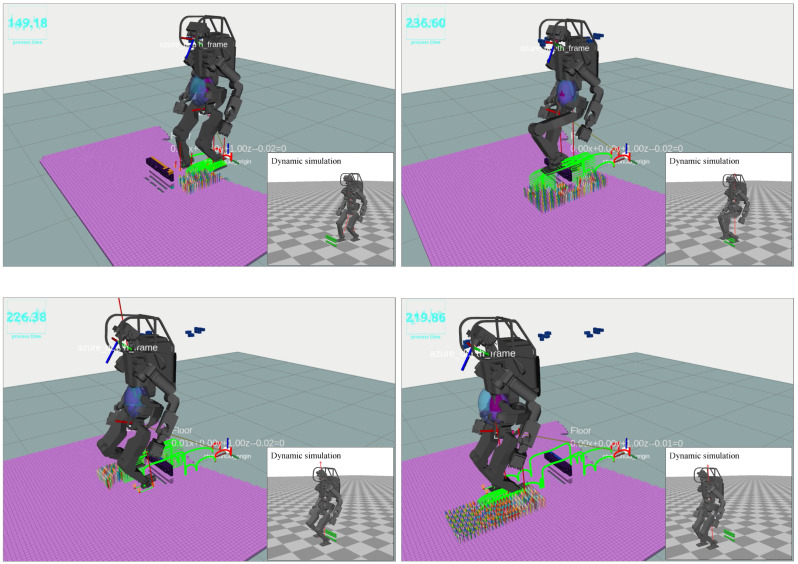
Surmounting a 20.0 cm hurdle. The reference motion data showed a basic, equidistant bipedal walking pattern premised on an entirely flat terrain. Leveraging a live data stream from the head camera to generate the ExOctomap, our planner delineates a collision–free path to adeptly navigate over the hurdle.

**Figure 17 sensors-23-08837-f017:**
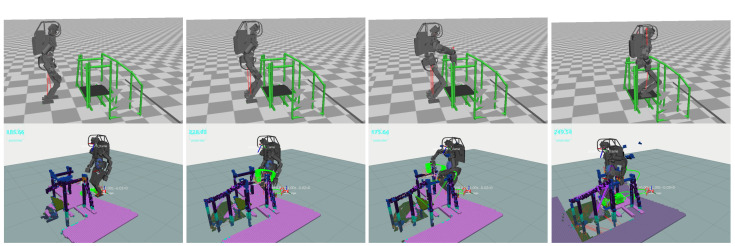
Navigating a scaffold mock-up via handrail grasping. The initial motion data use a basic bipedal walking pattern assuming a flat surface and a hand-grasping strategy with a 10 cm margin of error. As the robot moves, it expands the map progressively, allowing our planner to adjust contacts and devise collision-free trajectories adaptive to the real-time environment with each step.

**Figure 18 sensors-23-08837-f018:**
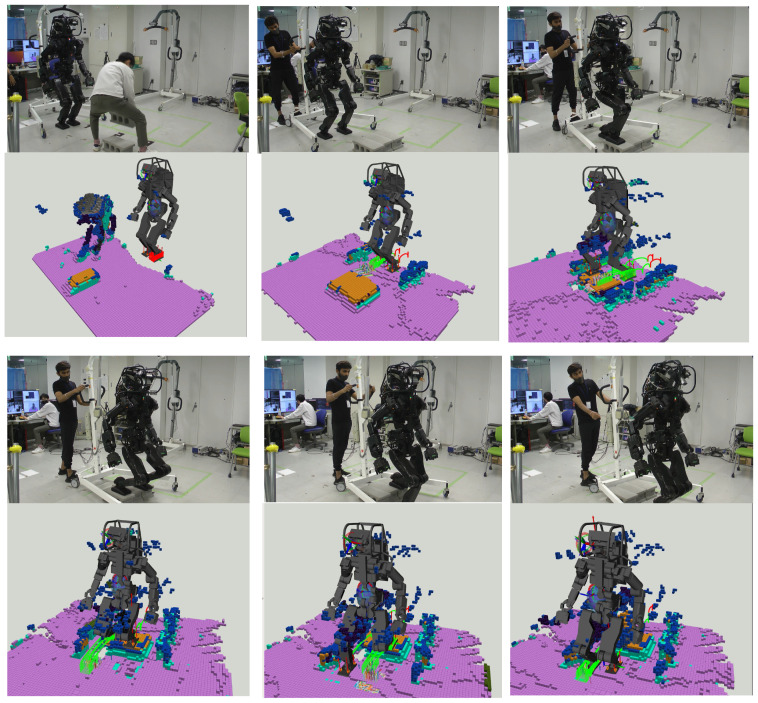
Walking over blocks: Initially, the robot possesses reference motion data comprised of flat foot sequences with 25.0 cm steps. When in operation, a person introduces blocks in its path, temporarily introducing the individual’s voxels to the ExOctomap, which are later removed through our ray-casting. As the robot encounters the obstacle, the planner generates climbing motions based on the ExOctomap. Image four highlights the planner’s capability to identify limited yet sufficient contact areas, enabling the robot to maintain its initial walking pattern substantially.

**Figure 19 sensors-23-08837-f019:**
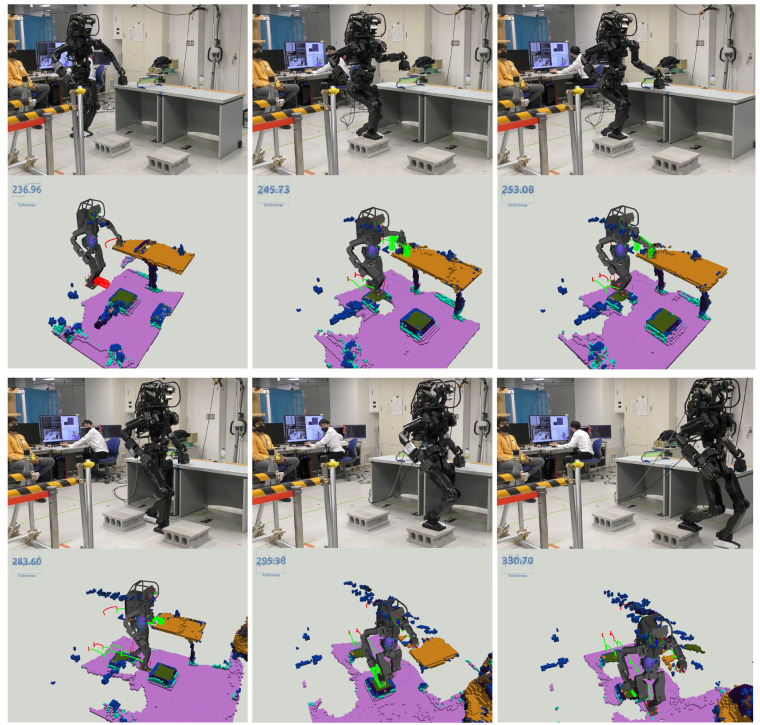
Navigating while interacting with tables: The robot initially had a recurring cycle involving a left foot, right foot, and left hand movement, each progressing by 25.0 cm. Our integrated vision and planning system dynamically adapts to each transition during the traversal, facilitating obstacle and hurdle navigation. The top-middle and top-right images vividly demonstrate the efficacy of the collision avoidance trajectory planning; recognizing a looming collision with a minor hurdle, the robot intelligently elevated its trajectory to avoid the obstacle.

## Data Availability

The data presented in this study are available on request from the corresponding author. The data are not publicly available due to the file size and privacy.

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
