# Peer review of "Online Multi-Contact Motion Replanning for Humanoid Robots with Semantic 3D Voxel Mapping: ExOctomap"

_sensors, 2023, doi:10.3390/s23218837_

Round 1

Reviewer 1 Report

Comments and Suggestions for Authors

Dear Authors,

The research work delves into a motion replanner against environmental uncertainties and incorporates novel techniques, such as the use of ExOctomap for voxel-based mapping and semantic features. The work is quite comprehensive and includes simulation and experimental trials. Very few observations need to be pointed out in this study.

Minor comments:

1. It is lightly suggested to drop the reference of Figure 1 in the abstract to provide the idea of how fast the planner performs, and it is recommended to provide the comparative baseline of the outcomes in the numerical analysis.

2.  It is suggested to avoid or include the term SLAM within the abstract or title of the work as it is a relevant term in the keywords.

3. The authors intend to provide an impact on the paper presenting a humanoid and challenging environment with uncertainty and dynamic properties. However, it should include in the text the real motivation of including Figure 1 at the very beginning of the document.  

4. It seems that Figure 2 misses the symbol $\times$ in the entire area.

5.  To the knowledge of the Reviewer and most, there is no information about QP controllers. Thus, it is suggested to cite it.

6. It is recommended a complete proofread of the entire document. There are some writing, typographical, and grammar mistakes, including subject-verb agreement, wording, punctuation, and symbolic mistakes.

Comments on the Quality of English Language

The authors have minor mistakes regarding punctuation, symbolic expressions, and syntax. However, it is quite well written and provides a good explanation of the replanning technique. 

Author Response

We have prepared a response letter to the reviewers in PDF format. Please see the attached file.

Reviewer 2 Report

Comments and Suggestions for Authors

The authors present a motion replanning algorithm, which utilizes an extended OctoMap (ExOctomap). The map is based on the well-known OctoMap data structure. It is augmented with additional semantic information, including normal vector, shape type, Eigen vectors and Eigen values. This map is utilized by the planner to effectively plan the motion of the robot. The system is evaluated in simulation as well as in the real world.

The investigated problem is of high interest for humanoid robotics. The paper is well written and well structured. The presented approach is sound. There are however a few points that need to be improved:

- Referring to figures in the abstract should be avoided, since the abstract may be shown in a context (e.g. a website), where the figure is not available.

- Somewhere at the beginning, a brief overview over the complete system (robot, localization/SLAM, mapping, planning, and control) should be given.

- You may consider to formulate Alg. 1 in terms of an actual procedure instead of a "while true" loop, e.g. having the current map, the most recent point cloud and the camera pose as input, and having the updated map as output.

- Sect. 3.2 should refer to the corresponding line in the algorithm. Furthermore, I suggest to move that section after 3.4 in order to match the order in the algorithm.

- The individual steps of Alg. 1 should be described more in detail. Furthermore, it is not stated explicitly which steps are executed on the most recent point cloud and which are executed on the whole map. I assume that lines 2-13 consider the point cloud, while lines 14-21 take place on the whole map. But this should be stated more clearly.

- In addition to cropping the input point cloud (line 5), do you also limit the area depicted by the whole map and move that window with the robot?

- In Sect. 5, all parameters of the algorithms used in the experiment should be given.

- The evaluation mainly neglects the ExOctomap. In particular, its quality as well as the computational and memory requirements should be investigated.

- How often do you update the ExOctomap?

- Are you planning to publish the proposed algorithm as open source project, in particular since you build up on other open source frameworks?

Comments on the Quality of English Language

Minor editing of English language required

Author Response

(The authors gave the same response as above.)

Reviewer 3 Report

Comments and Suggestions for Authors

This paper focuses on the robot motion replanning with semantic mapping. This article employs a combined approach of theory and practical validation to assess the feasibility of the algorithm. However, some minor revision should be made before publication.

1) When perceiving the surrounding environment with LiDAR on a robot, signal latency occurs during LiDAR signal processing. This latency phenomenon can result in reduced perception accuracy. Fortunately, the issue of signal latency has been addressed through the estimation-prediction framework in: secure cooperative localization for connected automated vehicles based on consensus, automated vehicle sideslip angle estimation considering signal measurement characteristic. The aforementioned work needs to be introduced and discussed.

2) For the mapping method, besides deep learning based method, Kalman filter with IMU and GPS fusion is also an alternative approach: integrated inertial-lidar-based map matching localization for varying environments. Based on the aforementioned work, please elaborate on the superiority of the method you have proposed.

3) The content of the conclusion is too lengthy and needs to be condensed and trimmed.

Author Response

(The authors gave the same response as above.)

Round 2

Reviewer 2 Report

Comments and Suggestions for Authors

The authors responded in detail to the reviewers' comments and addressed them carefully in the revised manuscript. In particular, several details and clarifications have been added, some sections have been restructured and reordered, algorithm 1 has been reformulated, and a comparison to the original OctoMap has been added. I've no further comments.

Comments on the Quality of English Language

Minor editing of English language required